# Benchmarking Neural Network Robustness to Common Corruptions and Perturbations

**Dan Hendrycks**
University of California, Berkeley
hendrycks@berkeley.edu

**Thomas Dietterich**
Oregon State University
tgd@oregonstate.edu

## Abstract

In this paper we establish rigorous benchmarks for image classifier robustness. Our first benchmark, IMAGENET-C, standardizes and expands the corruption robustness topic, while showing which classifiers are preferable in safety-critical applications. Then we propose a new dataset called IMAGENET-P which enables researchers to benchmark a classifier's robustness to common perturbations. Unlike recent robustness research, this benchmark evaluates performance on common corruptions and perturbations not worst-case adversarial perturbations. We find that there are negligible changes in relative corruption robustness from AlexNet classifiers to ResNet classifiers. Afterward we discover ways to enhance corruption and perturbation robustness. We even find that a bypassed adversarial defense provides substantial common perturbation robustness. Together our benchmarks may aid future work toward networks that robustly generalize.

## 1 Introduction

The human vision system is robust in ways that existing computer vision systems are not (Recht et al., 2018; Azulay & Weiss, 2018). Unlike current deep learning classifiers (Krizhevsky et al., 2012; He et al., 2015; Xie et al., 2016), the human vision system is not fooled by small changes in query images. Humans are also not confused by many forms of corruption such as snow, blur, pixelation, and novel combinations of these. Humans can even deal with abstract changes in structure and style. Achieving these kinds of robustness is an important goal for computer vision and machine learning. It is also essential for creating deep learning systems that can be deployed in safety-critical applications.

Most work on robustness in deep learning methods for vision has focused on the important challenges of robustness to adversarial examples (Szegedy et al., 2014; Carlini & Wagner, 2017; 2016), unknown unknowns (Hendrycks et al., 2019; Hendrycks & Gimpel, 2017b; Liu et al., 2018), and model or data poisoning (Steinhardt et al., 2017; Hendrycks et al., 2018). In contrast, we develop and validate datasets for two other forms of robustness. Specifically, we introduce the IMAGETNET-C dataset for input *corruption robustness* and the IMAGENET-P dataset for input *perturbation robustness*.

To create IMAGENET-C, we introduce a set of 75 common visual corruptions and apply them to the ImageNet object recognition challenge (Deng et al., 2009). We hope that this will serve as a general dataset for benchmarking robustness to image corruptions and prevent methodological problems such as moving goal posts and result cherry picking. We evaluate the performance of current deep learning systems and show that there is wide room for improvement on IMAGENET-C. We also introduce a total of three methods and architectures that improve corruption robustness without losing accuracy.

To create IMAGENET-P, we introduce a set of perturbed or subtly differing ImageNet images. Using metrics we propose, we measure the stability of the network's predictions on these perturbed images. Although these perturbations are not chosen by an adversary, currently existing networks exhibit surprising instability on common perturbations. Then we then demonstrate that approaches which enhance corruption robustness can also improve perturbation robustness. For example, some recent architectures can greatly improve both types of robustness. More, we show that the Adversarial Logit Pairing $\ell_\infty$ adversarial example defense can yield substantial robustness gains on diverse and common perturbations. By defining and benchmarking perturbation and corruption robustness, we facilitate research that can be overcome by future networks which do not rely on spurious correlations or cues inessential to the object's class.

## 2 RELATED WORK

**Adversarial Examples.** An adversarial image is a clean image perturbed by a small distortion carefully crafted to confuse a classifier. These deceptive distortions can occasionally fool black-box classifiers (Kurakin et al., 2017). Algorithms have been developed that search for the smallest additive distortions in RGB space that are sufficient to confuse a classifier (Carlini et al., 2017). Thus adversarial distortions serve as type of worst-case analysis for network robustness. Its popularity has often led "adversarial robustness" to become interchangeable with "robustness" in the literature (Bastani et al., 2016; Rauber et al., 2017). In the literature, new defenses (Lu et al., 2017; Papernot et al., 2017; Metzen et al., 2017; Hendrycks & Gimpel, 2017a) often quickly succumb to new attacks (Evtimov et al., 2017; Carlini & Wagner, 2017; 2016), with some exceptions for perturbations on small images (Schott et al., 2018; Madry et al., 2018). For some simple datasets, the existence of any classification error ensures the existence of adversarial perturbations of size $\mathcal{O}(d^{-1/2})$, $d$ the input dimensionality (Gilmer et al., 2018b). For some simple models, adversarial robustness requires an increase in the training set size that is polynomial in $d$ (Schmidt et al., 2018). Gilmer et al. (2018a) suggest modifying the problem of adversarial robustness itself for increased real-world applicability.

**Robustness in Speech.** Speech recognition research emphasizes robustness to common corruptions rather than worst-case, adversarial corruptions (Li et al., 2014; Mitra et al., 2017). Common acoustic corruptions (e.g., street noise, background chatter, wind) receive greater focus than adversarial audio, because common corruptions are ever-present and unsolved. There are several popular datasets containing noisy test audio (Hirsch & Pearce, 2000; Hirsch, 2007). Robustness in noisy environments requires robust architectures, and some research finds convolutional networks more robust than fully connected networks (Abdel-Hamid et al., 2013). Additional robustness has been achieved through pre-processing techniques such as standardizing the statistics of the input (Liu et al., 1993; Torre et al., 2005; Harvilla & Stern, 2012; Kim & Stern, 2016).

**ConvNet Fragility Studies.** Several studies demonstrate the fragility of convolutional networks on simple corruptions. For example, Hosseini et al. (2017) apply impulse noise to break Google's Cloud Vision API. Using Gaussian noise and blur, Dodge & Karam (2017b) demonstrate the superior robustness of human vision to convolutional networks, *even after networks are fine-tuned* on Gaussian noise or blur. Geirhos et al. (2017) compare networks to humans on noisy and elastically deformed images. They find that fine-tuning on specific corruptions does not generalize and that classification error patterns underlying network and human predictions are not similar. Temel et al. (2017; 2018); Temel & AlRegib (2018) propose different corrupted datasets for object and traffic sign recognition.

**Robustness Enhancements.** In an effort to reduce classifier fragility, Vasiljevic et al. (2016) fine-tune on blurred images. They find it is not enough to fine-tune on one type of blur to generalize to other blurs. Furthermore, fine-tuning on several blurs can marginally decrease performance. Zheng et al. (2016) also find that fine-tuning on noisy images can cause underfitting, so they encourage the noisy image softmax distribution to match the clean image softmax. Dodge & Karam (2017a) address underfitting via a mixture of corruption-specific experts assuming corruptions are known beforehand.

## 3 CORRUPTIONS, PERTURBATIONS, AND ADVERSARIAL PERTURBATIONS

We now define corruption and perturbation robustness and distinguish them from adversarial perturbation robustness. To begin, we consider a classifier $f : \mathcal{X} \rightarrow \mathcal{Y}$ trained on samples from distribution $\mathcal{D}$, a set of corruption functions $C$, and a set of perturbation functions $\mathcal{E}$. We let $\mathbb{P}_C(c), \mathbb{P}_{\mathcal{E}}(\varepsilon)$ approximate the real-world frequency of these corruptions and perturbations. Most classifiers are judged by their accuracy on test queries drawn from $\mathcal{D}$, i.e., $\mathbb{P}_{(x,y)\sim\mathcal{D}}(f(x) = y)$. Yet in a vast range of cases the classifier is tasked with classifying low-quality or corrupted inputs. In view of this, we suggest also computing the classifier's *corruption robustness* $\mathbb{E}_{c\sim C}[\mathbb{P}_{(x,y)\sim\mathcal{D}}(f(c(x) = y))]$. This contrasts with a popular notion of adversarial robustness, often formulated $\min_{\|\delta\|_p < b} \mathbb{P}_{(x,y)\sim\mathcal{D}}(f(x + \delta) = y)$, $b$ a small budget. Thus, corruption robustness measures the classifier's average-case performance on corruptions $C$, while adversarial robustness measures the worst-case performance on small, additive, classifier-tailored perturbations.

Average-case performance on small, general, classifier-agnostic perturbations motivates us to define *perturbation robustness*, namely $\mathbb{E}_{\varepsilon\sim\mathcal{E}}[\mathbb{P}_{(x,y)\sim\mathcal{D}}(f(\varepsilon(x)) = f(x))]$. Consequently, in measuring perturbation robustness, we track the classifier's prediction stability, reliability, or consistency in the

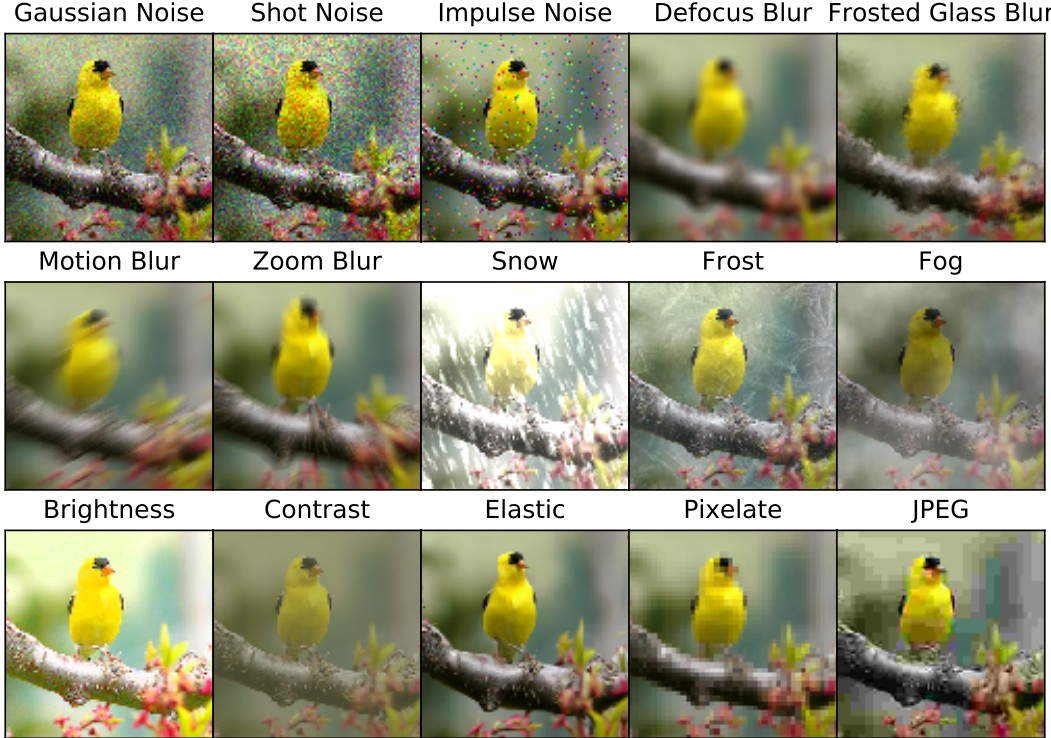

Figure 1: Our IMAGENET-C dataset consists of 15 types of algorithmically generated corruptions from noise, blur, weather, and digital categories. Each type of corruption has five levels of severity, resulting in 75 distinct corruptions. See different severity levels in Appendix B.

face of minor input changes. Now in order to approximate $C, \mathcal{E}$ and these robustness measures, we designed a set of corruptions and perturbations which are frequently encountered in natural images. We will refer to these as "common" corruptions and perturbations. These common corruptions and perturbations are available in the form of IMAGENET-C and IMAGENET-P.

## 4 THE IMAGENET-C AND IMAGENET-P ROBUSTNESS BENCHMARKS

### 4.1 THE DATA OF IMAGENET-C AND IMAGENET-P

**IMAGENET-C Design.** The IMAGENET-C benchmark consists of 15 diverse corruption types applied to validation images of ImageNet. The corruptions are drawn from four main categories—noise, blur, weather, and digital—as shown in Figure 1. Research that improves performance on this benchmark should indicate general robustness gains, as the corruptions are diverse and numerous. Each corruption type has five levels of severity since corruptions can manifest themselves at varying intensities. Appendix A gives an example of the five different severity levels for impulse noise. Real-world corruptions also have variation even at a fixed intensity. To simulate these, we introduce variation for each corruption when possible. For example, each fog cloud is unique to each image. These algorithmically generated corruptions are applied to the ImageNet (Deng et al., 2009) validation images to produce our corruption robustness dataset IMAGENET-C. The dataset can be downloaded or re-created by visiting https://github.com/hendrycks/robustness. IMAGENET-C images are saved as lightly compressed JPEGs; this implies an image corrupted by Gaussian noise is also slightly corrupted by JPEG compression. Our benchmark tests networks with IMAGENET-C images, *but networks should not be trained on these images*. Networks should be trained on datasets such as ImageNet and not be trained on IMAGENET-C corruptions. To enable further experimentation, we designed an extra corruption type for each corruption category (Appendix B), and we provide CIFAR-10-C, TINY IMAGENET-C, IMAGENET $64 \times 64$-C, and Inception-sized editions. Overall, the IMAGENET-C dataset consists of 75 corruptions, all applied to ImageNet validation images for testing a pre-existing network.

**Common Corruptions.**    The first corruption type is *Gaussian noise*. This corruption can appear in low-lighting conditions. *Shot noise*, also called Poisson noise, is electronic noise caused by the discrete nature of light itself. *Impulse noise* is a color analogue of salt-and-pepper noise and can be caused by bit errors. *Defocus blur* occurs when an image is out of focus. *Frosted Glass Blur* appears with "frosted glass" windows or panels. *Motion blur* appears when a camera is moving quickly. *Zoom blur* occurs when a camera moves toward an object rapidly. *Snow* is a visually obstructive form of precipitation. *Frost* forms when lenses or windows are coated with ice crystals. *Fog* shrouds objects and is rendered with the diamond-square algorithm. *Brightness* varies with daylight intensity. *Contrast* can be high or low depending on lighting conditions and the photographed object's color. *Elastic* transformations stretch or contract small image regions. *Pixelation* occurs when upsampling a low-resolution image. *JPEG* is a lossy image compression format which introduces compression artifacts.

**IMAGENET-P Design.**    The second benchmark that we propose tests the classifier's perturbation robustness. Models lacking in perturbation robustness produce erratic predictions which undermines user trust. When perturbations have a high propensity to change the model's response, then perturbations could also misdirect or destabilize iterative image optimization procedures appearing in style transfer (Gatys et al., 2016), decision explanations (Fong & Vedaldi, 2017), feature visualization (Olah et al., 2017), and so on. Like IMAGENET-C, IMAGENET-P consists of noise, blur, weather, and digital distortions. Also as before, the dataset has validation perturbations; has difficulty levels; has CIFAR-10, Tiny ImageNet, ImageNet $64 \times 64$, standard, and Inception-sized editions; and has been designed for benchmarking not training networks. IMAGENET-P departs from IMAGENET-C by having perturbation sequences generated from each ImageNet validation image; examples are in Figure 2. Each sequence contains more than 30 frames, so we counteract an increase in dataset size and evaluation time by using only 10 common perturbations.

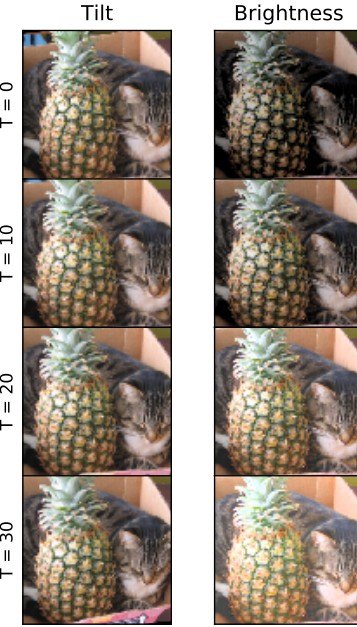

Figure 2: Example frames from the beginning ($T = 0$) to end ($T = 30$) of some Tilt and Brightness perturbation sequences.

**Common Perturbations.**    Appearing more subtly than the corruption from IMAGENET-C, the Gaussian noise perturbation sequence begins with the clean ImageNet image. The following frames in the sequence consist in the same image but with minute Gaussian noise perturbations applied. This sequence design is similar for the shot noise perturbation sequence. However the remaining perturbation sequences have temporality, so that each frame of the sequence is a perturbation of the previous frame. Since each perturbation is small, repeated application of a perturbation does not bring the image far out-of-distribution. For example, an IMAGENET-P translation perturbation sequence shows a clean ImageNet image sliding from right to left one pixel at a time; with each perturbation of the pixel locations, the resulting frame is still of high quality. The perturbation sequences with temporality are created with motion blur, zoom blur, snow, brightness, translate, rotate, tilt (viewpoint variation through minor 3D rotations), and scale perturbations.

## 4.2    IMAGENET-C AND IMAGENET-P METRICS AND SETUP

**IMAGENET-C Metrics.**    Common corruptions such as Gaussian noise can be benign or destructive depending on their severity. In order to *comprehensively* evaluate a classifier's robustness to a given type of corruption, we score the classifier's performance across five corruption severity levels and aggregate these scores. The first evaluation step is to take a trained classifier f, which has not been trained on IMAGENET-C, and compute the clean dataset top-1 error rate. Denote this error rate $E_{\text{clean}}^f$. The second step is to test the classifier on each corruption type $c$ at each level of severity $s$ ($1 \le s \le 5$). This top-1 error is written $E_{s,c}^f$. Before we aggregate the classifier's performance across severities and corruption types, we will make error rates more comparable since different corruptions pose different levels of difficulty. For example, fog corruptions often obscure an object's class more than brightness corruptions. We adjust for the varying difficulties by dividing by AlexNet's errors,

but any baseline will do (even a baseline with 100% error rates, corresponding to an average of CEs). This standardized aggregate performance measure is the Corruption Error, computed with the formula

$$\text{CE}_c^f = \left( \sum_{s=1}^{5} E_{s,c}^f \right) \Big/ \left( \sum_{s=1}^{5} E_{s,c}^{\text{AlexNet}} \right).$$

Now we can summarize model corruption robustness by averaging the 15 Corruption Error values $\text{CE}_{\text{Gaussian Noise}}^f, \text{CE}_{\text{Shot Noise}}^f, \ldots, \text{CE}_{\text{JPEG}}^f$. This results in the *mean CE* or *mCE* for short.

We now introduce a more nuanced corruption robustness measure. Consider a classifier that withstands most corruptions, so that the gap between the mCE and the clean data error is minuscule. Contrast this with a classifier with a low clean error rate which has its error rate spike in the presence of corruptions; this corresponds to a large gap between the mCE and clean data error. It is possible that the former classifier has a larger mCE than the latter, despite the former degrading more gracefully in the presence of corruptions. The amount that the classifier declines on corrupted inputs is given by the formula Relative $\text{CE}_c^f = \left( \sum_{s=1}^{5} E_{s,c}^f - E_{\text{clean}}^f \right) \big/ \left( \sum_{s=1}^{5} E_{s,c}^{\text{AlexNet}} - E_{\text{clean}}^{\text{AlexNet}} \right)$. Averaging these 15 Relative Corruption Errors results in the *Relative mCE*. This measures the relative robustness or the performance degradation when encountering corruptions.

**IMAGENET-P Metrics.** A straightforward approach to estimate $\mathbb{E}_{\varepsilon \sim \mathcal{E}}[\mathbb{P}_{(x,y) \sim \mathcal{D}}(f(\varepsilon(x)) \neq f(x))]$ falls into place when using IMAGENET-P perturbation sequences. Let us denote $m$ perturbation sequences with $\mathcal{S} = \left\{ \left( x_1^{(i)}, x_2^{(i)}, \ldots, x_n^{(i)} \right) \right\}_{i=1}^{m}$ where each sequence is made with perturbation $p$. The "Flip Probability" of network $f : \mathcal{X} \to \{1, 2, \ldots, 1000\}$ on perturbation sequences $\mathcal{S}$ is

$$\text{FP}_p^f = \frac{1}{m(n-1)} \sum_{i=1}^{m} \sum_{j=2}^{n} \mathbb{1}\big(f\big(x_j^{(i)}\big) \neq f\big(x_{j-1}^{(i)}\big)\big) = \mathbb{P}_{x \sim \mathcal{S}}(f(x_j) \neq f(x_{j-1})).$$

For noise perturbation sequences, which are not temporally related, $x_1^{(i)}$ is clean and $x_j^{(i)}$ ($j > 1$) are perturbed images of $x_1^{(i)}$. We can recast the FP formula for noise sequences as $\text{FP}_p^f = \frac{1}{m(n-1)} \sum_{i=1}^{m} \sum_{j=2}^{n} \mathbb{1}\big(f\big(x_j^{(i)}\big) \neq f\big(x_1^{(i)}\big)\big) = \mathbb{P}_{x \sim \mathcal{S}}(f(x_j) \neq f(x_1) \mid j > 1)$. As was done with the Corruption Error formula, we now standardize the Flip Probability by the sequence's difficulty for increased commensurability. We have, then, the "Flip Rate" $\text{FR}_p^f = \text{FP}_p^f / \text{FP}_p^{\text{AlexNet}}$. Averaging the Flip Rate across all perturbations yields the *mean Flip Rate* or *mFR*. We do not define a "relative mFR" since we did not find any natural formulation, nor do we directly use predicted class probabilities due to differences in model calibration (Guo et al., 2017).

When the top-5 predictions are relevant, perturbations should not cause the list of top-5 predictions to shuffle chaotically, nor should classes sporadically vanish from the list. We penalize top-5 inconsistency of this kind with a different measure. Let the ranked predictions of network $f$ on $x$ be the permutation $\tau(x) \in S_{1000}$. Concretely, if "Toucan" has the label 97 in the output space and "Pelican" has the label 145, and if $f$ on $x$ predicts "Toucan" and "Pelican" to be the most and second-most likely classes, respectively, then $\tau(x)(97) = 1$ and $\tau(x)(144) = 2$. These permutations contain the top-5 predictions, so we use permutations to compare top-5 lists. To do this, we define

$$d(\tau(x), \tau(x')) = \sum_{i=1}^{5} \sum_{j=\min\{i,\sigma(i)\}+1}^{\max\{i,\sigma(i)\}} \mathbb{1}(1 \leq j - 1 \leq 5)$$

where $\sigma = (\tau(x))^{-1}\tau(x')$. If the top-5 predictions represented within $\tau(x)$ and $\tau(x')$ are identical, then $d(\tau(x), \tau(x')) = 0$. More examples of $d$ on several permutations are in Appendix C. Comparing the top-5 predictions across entire perturbation sequences results in the unstandardized Top-5 Distance $\text{uT5D}_p^f = \frac{1}{m(n-1)} \sum_{i=1}^{m} \sum_{j=2}^{n} d(\tau(x_j), \tau(x_{j-1})) = \mathbb{P}_{x \sim \mathcal{S}}(d(\tau(x_j), \tau(x_{j-1}))$. For noise perturbation sequences, we have $\text{uT5D}_p^f = \mathbb{E}_{x \sim \mathcal{S}}[d(\tau(x_j), \tau(x_1)) \mid j > 1]$. Once the uT5D is standardized, we have the Top-5 Distance $\text{T5D}_p^f = \text{uT5D}_p^f / \text{uT5D}_p^{\text{AlexNet}}$. The T5Ds averaged together correspond to the *mean Top-5 Distance* or *mT5D*.

**Preserving Metric Validity.** The goal of IMAGENET-C and IMAGENET-P is to evaluate the robustness of machine learning algorithms on novel corruptions and perturbations. Humans are able to generalize to novel corruptions quite well; for example, they can easily deal with new Instagram filters. Likewise for perturbations; humans relaxing in front of an undulating ocean do not give turbulent ac-

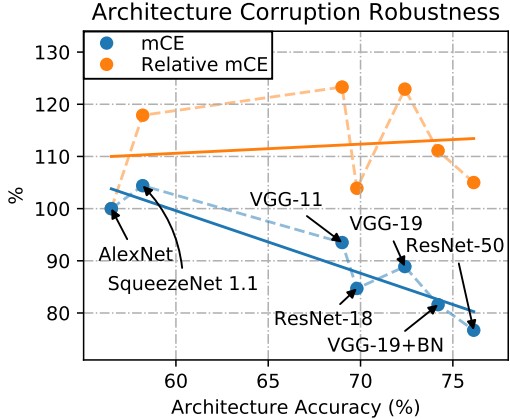
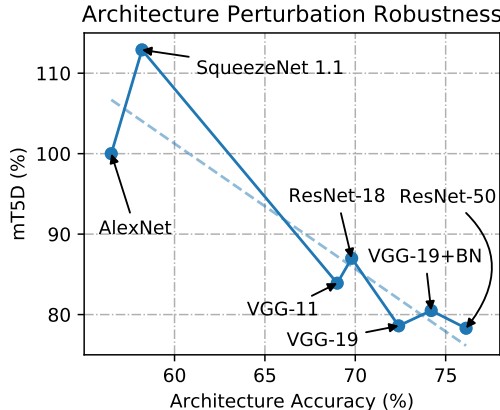

Figure 3: Robustness (mCE) and Relative mCE IMAGENET-C values. Relative mCE values suggest robustness in itself declined from AlexNet to ResNet. "BN" abbreviates Batch Normalization.

Figure 4: Perturbation robustness of various architectures as measured by the mT5D on IMAGENET-P. Observe that corruption and perturbation robustness track distinct concepts.

| | | | Noise | | | Blur | | | | Weather | | | | Digital | | | |
|---|---|---|---|---|---|---|---|---|---|---|---|---|---|---|---|---|---|
| Network | Error | mCE | Gauss. | Shot | Impulse | Defocus | Glass | Motion | Zoom | Snow | Frost | Fog | Bright | Contrast | Elastic | Pixel | JPEG |
| AlexNet | 43.5 | 100.0 | 100 | 100 | 100 | 100 | 100 | 100 | 100 | 100 | 100 | 100 | 100 | 100 | 100 | 100 | 100 |
| SqueezeNet | 41.8 | 104.4 | 107 | 106 | 105 | 100 | 103 | 101 | 100 | 101 | 103 | 97 | 97 | 98 | 106 | 109 | 134 |
| VGG-11 | 31.0 | 93.5 | 97 | 97 | 100 | 92 | 99 | 93 | 91 | 92 | 91 | 84 | 75 | 86 | 97 | 107 | 100 |
| VGG-19 | 27.6 | 88.9 | 89 | 91 | 95 | 89 | 98 | 90 | 90 | 89 | 86 | 75 | 68 | 80 | 97 | 102 | 94 |
| VGG-19+BN | 25.8 | 81.6 | 82 | 83 | 88 | 82 | 94 | 84 | 86 | 80 | 78 | 69 | 61 | 74 | 94 | 85 | 83 |
| ResNet-18 | 30.2 | 84.7 | 87 | 88 | 91 | 84 | 91 | 87 | 89 | 86 | 84 | 78 | 69 | 78 | 90 | 80 | 85 |
| ResNet-50 | 23.9 | 76.7 | 80 | 82 | 83 | 75 | 89 | 78 | 80 | 78 | 75 | 66 | 57 | 71 | 85 | 77 | 77 |

Table 1: Clean Error, mCE, and Corruption Error values of different corruptions and architectures on IMAGENET-C. The mCE value is the mean Corruption Error of the corruptions in Noise, Blur, Weather, and Digital columns. Models are trained only on clean ImageNet images.

counts of the scenery before them. Hence, we propose the following protocol. The image recognition network should be trained on the ImageNet training set and on whatever other training sets the investigator wishes to include. Researchers should clearly state whether they trained on these corruptions or perturbations; however, this training strategy is discouraged (see Section 2). We allow training with other distortions (e.g., uniform noise) and standard data augmentation (i.e., cropping, mirroring), even though cropping overlaps with translations. Then the resulting trained model should be evaluated on IMAGENET-C or IMAGENET-P using the above metrics. Optionally, researchers can test with the separate set of validation corruptions and perturbations we provide for IMAGENET-C and IMAGENET-P.

## 5 EXPERIMENTS

### 5.1 ARCHITECTURE ROBUSTNESS

How robust are current methods, and has progress in computer vision been achieved at the expense of robustness? As seen in Figure 3, as architectures improve, so too does the mean Corruption Error (mCE). By this measure, architectures have become progressively more successful at generalizing to corrupted distributions. Note that models with similar clean error rates have fairly similar CEs, and in Table 1 there are no large shifts in a corruption type's CE. Consequently, it would seem that architectures have slowly and consistently improved their representations over time. However, it appears that corruption robustness improvements are mostly explained by accuracy improvements. Recall that the Relative mCE tracks a classifier's accuracy *decline* in the presence of corruptions. Figure 3 shows that the Relative mCEs of many subsequent models are worse than that of AlexNet (Krizhevsky et al., 2012). Full results are in Appendix D. In consequence, from AlexNet to ResNet, corruption robustness in itself has barely changed. Thus our "superhuman" classifiers are decidedly subhuman.

On perturbed inputs, current classifiers are unexpectedly bad. For example, a ResNet-18 on Scale perturbation sequences have a 15.6% probability of flipping its top-1 prediction between adjacent

frames (i.e., $\text{FP}^{\text{ResNet-18}}_{\text{Scale}} = 15.6\%$); the $\text{uT5D}^{\text{ResNet-18}}_{\text{Scale}}$ is 3.6. More results are in Appendix E. Clearly perturbations need not be adversarial to fool current classifiers. What is also surprising is that while VGGNets are worse than ResNets at generalizing to corrupted examples, on perturbed examples they can be just as robust or even more robust. Likewise, Batch Normalization made VGG-19 less robust to perturbations but more robust to corruptions. Yet this is not to suggest that there is a fundamental trade-off between corruption and perturbation robustness. In fact, both corruption and perturbation robustness can improve together, as we shall see later.

## 5.2   ROBUSTNESS ENHANCEMENTS

Be aware that Appendix F contains many informative failures in robustness enhancement. Those experiments underscore the necessity in testing on a a diverse test set, the difficulty in cleansing corruptions from image, and the futility in expecting robustness gains from some "simpler" models.

**Histogram Equalization.**   Histogram equalization successfully standardizes speech data for robust speech recognition (Torre et al., 2005; Harvilla & Stern, 2012). For images, we find that preprocessing with Contrast Limited Adaptive Histogram Equalization (Pizer et al., 1987) is quite effective. Unlike our image denoising attempt (Appendix F), CLAHE reduces the effect of some corruptions while not worsening performance on most others, thereby improving the mCE. We demonstrate CLAHE's net improvement by taking a pre-trained ResNet-50 and fine-tuning the whole model for five epochs on images processed with CLAHE. The ResNet-50 has a 23.87% error rate, but ResNet-50 with CLAHE has an error rate of 23.55%. On nearly all corruptions, CLAHE slightly decreases the Corruption Error. The ResNet-50 without CLAHE preprocessing has an mCE of 76.7%, while with CLAHE the ResNet-50's mCE decreases to 74.5%.

**Multiscale Networks.**   Multiscale architectures achieve greater corruption robustness by propagating features across scales at each layer rather than slowly gaining a global representation of the input as in typical convolutional neural networks. Some multiscale architectures are called Multigrid Networks (Ke et al., 2017). Multigrid networks each have a pyramid of grids in each layer which enables the subsequent layer to operate across scales. Along similar lines, Multi-Scale Dense Networks (MSDNets) (Huang et al., 2018) use information across scales. MSDNets bind network layers with DenseNet-like (Huang et al., 2017b) skip connections. These two different multiscale networks both enhance corruption robustness, but they do not provide any noticeable benefit in perturbation robustness. Now before comparing mCE values, we first note the Multigrid network has a 24.6% top-1 error rate, as does the MSDNet, while the ResNet-50 has a 23.9% top-1 error rate. On noisy inputs, Multigrid networks noticeably surpass ResNets and MSDNets, as shown in Figure 5. Since multiscale architectures have high-level representations processed in tandem with fine details, the architectures appear better equipped to suppress otherwise distracting pixel noise. When all corruptions are evaluated, ResNet-50 has an mCE of 76.7%, the MSDNet has an mCE of 73.6%, and the Multigrid network has an mCE of 73.3%.

**Feature Aggregating and Larger Networks.**   Some recent models enhance the ResNet architecture by increasing what is called feature aggregation. Of these, DenseNets and ResNeXts (Xie et al., 2016) are most prominent. Each purports to have stronger representations than ResNets, and the evidence is largely a hard-won ImageNet error-rate downtick. Interestingly, the IMAGENET-C mCE clearly indicates that DenseNets and ResNeXts have superior representations. Accordingly, a switch from a ResNet-50 (23.9% top-1 error) to a DenseNet-121 (25.6% error) decreases the mCE from 76.7% to 73.4% (and the relative mCE from 105.0% to 92.8%). More starkly, switching from a ResNet-50 to a ResNeXt-50 (22.9% top-1) drops the mCE from *76.7% to 68.2%* (relative mCE decreases from 105.0% to 88.6%). Corruption robustness results are summarized in Figure 5. This shows that corruption robustness may be a better way to measure future progress in representation learning than the clean dataset top-1 error rate.

Some of the greatest and simplest robustness gains sometimes emerge from making recent models more monolithic. Apparently more representations, more redundancy, and more capacity allow these massive models to operate more stably on corrupted inputs. We saw earlier that making models smaller does the opposite. Swapping a DenseNet-121 (25.6% top-1) with the larger DenseNet-161 (22.9% top-1) decreases the mCE from 73.4% to 66.4% (and the relative mCE from 92.8% to 84.6%). In a similar fashion, a ResNeXt-50 (22.9% top-1) is less robust than the a giant ResNeXt-101 (21.0% top-1). The mCEs are 68.2% and 62.2% respectively (and the relative mCEs are 88.6% and 80.1% respectively).

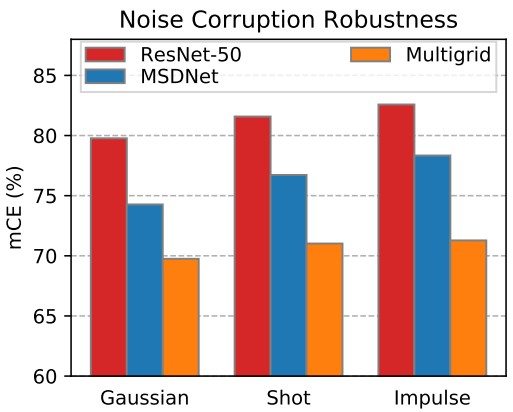
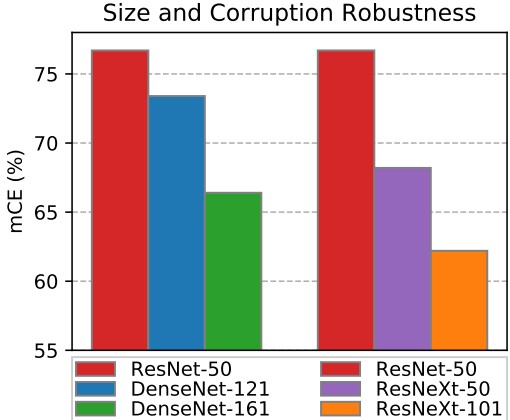

Figure 5: Architectures such as Multigrid networks and DenseNets resist noise corruptions more effectively than ResNets.

Figure 6: Larger feature aggregating networks achieve robustness gains that substantially outpace their accuracy gains.

Both model size and feature aggregation results are summarized in Figure 6. Consequently, future models with even more depth, width, and feature aggregation may attain further corruption robustness.

Feature aggregation and their larger counterparts similarly improve perturbation robustness. While a ResNet-50 has a 58.0% mFR and a 78.3% mT5D, a DenseNet-121 obtains a 56.4% mFR and 76.8% mT5D, and a ResNeXt-50 does even better with a 52.4% mFR and a 74.2% mT5D. Reflecting the corruption robustness findings further, the larger DenseNet-161 has a 46.9% mFR and 69.5% mT5D, while the ResNeXt-101 has a 43.2% mFR and 65.9% mT5D. Thus in two senses feature aggregating networks and their larger versions markedly enhance robustness.

**Stylized ImageNet.** Geirhos et al. (2019) propose a novel data augmentation scheme where ImageNet images are stylized with style transfer. The intent is that classifiers trained on stylized images will rely less on textural cues for classification. When a ResNet-50 is trained on typical ImageNet images and stylized ImageNet images, the resulting model has an mCE of 69.3%, down from 76.7%.

**Adversarial Logit Pairing.** ALP is an adversarial example defense for large-scale image classifiers (Kannan et al., 2018). Like nearly all other adversarial defenses, ALP was bypassed and has unclear value as an adversarial defense going forward (Engstrom et al., 2018), yet this is not a decisive reason dismiss it. ALP provides significant perturbation robustness even though it does not provide much adversarial perturbation robustness against all adversaries. Although ALP was designed to increase robustness to small gradient perturbations, it markedly improves robustness to all sorts of noise, blur, weather, and digital IMAGENET-P perturbations—methods generalizing this well is a rarity. In point of fact, a publicly available Tiny ImageNet ResNet-50 model fine-tuned with ALP has a 41% and 40% relative decrease in the mFP and mT5D on TINY IMAGENET-P, respectively. ALP's success in enhancing common perturbation robustness and its modest utility for adversarial perturbation robustness highlights that the interplay between these problems should be better understood.

# 6  CONCLUSION

In this paper, we introduced what are to our knowledge the first comprehensive benchmarks for corruption and perturbation robustness. This was made possible by introducing two new datasets, IMAGENET-C and IMAGENET-P. The first of which showed that many years of architectural advancements corresponded to minuscule changes in relative corruption robustness. Therefore benchmarking and improving robustness deserves attention, especially as top-1 clean ImageNet accuracy nears its ceiling. We also saw that classifiers exhibit unexpected instability on simple perturbations. Thereafter we found that methods such as histogram equalization, multiscale architectures, and larger feature-aggregating models improve corruption robustness. These larger models also improve perturbation robustness. However, we found that even greater perturbation robustness can come from an adversarial defense designed for adversarial $\ell_\infty$ perturbations, indicating a surprising interaction between adversarial and common perturbation robustness. In this work, we found several methods to increase robustness, introduced novel experiments and metrics, and created new datasets for the rigorous study of model robustness, a pressing necessity as models are unleashed into safety-critical real-world settings.

## 7 ACKNOWLEDGEMENTS

We should like to thank Justin Gilmer, David Wagner, Kevin Gimpel, Tom Brown, Mantas Mazeika, and Steven Basart for their helpful suggestions. This research was supported by a grant from the Future of Life Institute.

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

## A    EXAMPLE OF IMAGENET-C SEVERITIES

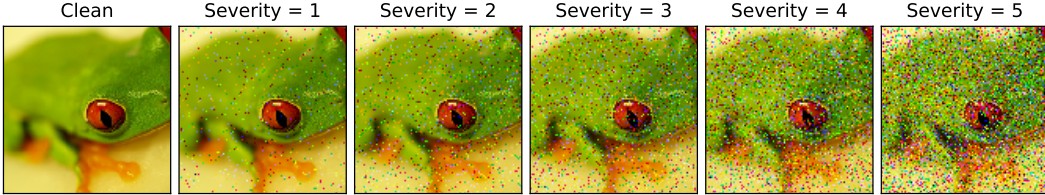

Figure 7: Impulse noise modestly to markedly corrupts a frog, showing our benchmark's varying severities.

In Figure 7, we show the Impulse noise corruption type in five different severities. Clearly, IMAGENET-C corruptions can range from negligible to pulverizing. Because of this range, the benchmark comprehensively assesses each corruption type.

## B    EXTRA IMAGENET-C CORRUPTIONS

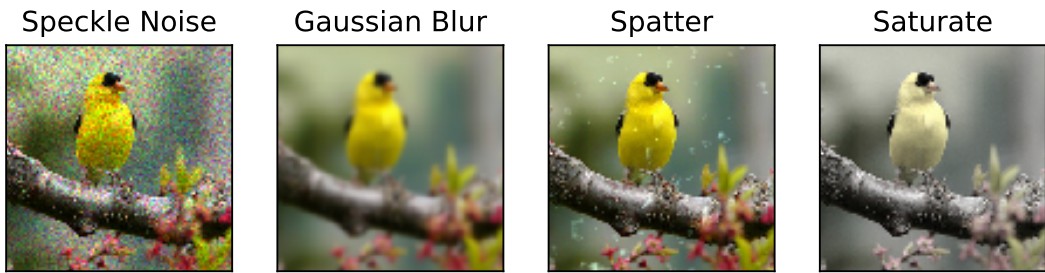

Figure 8: Extra IMAGENET-C corruption examples are available for model validation and sounder experimentation.

Directly fitting the types of IMAGENET-C corruptions should be avoided, as it would cause researchers to overestimate a model's robustness. Therefore, it is incumbent on us to simplify model validation. This is why we provide an additional form of corruption for each of the four general types. These are available for download at https://github.com/hendrycks/robustness. There is one corruption type for each noise, blur, weather, and digital category in the validation set. The first corruption type is *speckle noise*, an additive noise where the noise added to a pixel tends to be larger if the original pixel intensity is larger. *Gaussian blur* is a low-pass filter where a blurred pixel is a result of a weighted average of its neighbors, and farther pixels have decreasing weight in this average. *Spatter* can occlude a lens in the form of rain or mud. Finally, *saturate* is common in edited images where images are made more or less colorful. See Figure 8 for instances of each corruption type.

## C    MORE ON THE IMAGENET-P METRICS AND SETUP

For some readers, the following function may be opaque,

$$d(\tau(x), \tau(x')) = \sum_{i=1}^{5} \sum_{j=\min\{i,\sigma(i)\}+1}^{\max\{i,\sigma(i)\}} \mathbb{1}(1 \leq j - 1 \leq 5)$$

where $\sigma = (\tau(x))^{-1}\tau(x')$ and the empty sum is understood to be zero. A high-level view of $d$ is that it computes the deviation between the top-5 predictions of two prediction lists. For simplicity we find the deviation between the identity and $\sigma$ rather than $\tau(x)$ and $\tau(x')$. In consequence we can consider $d'(\sigma) := d(1, \sigma)$ where 1 the identity permutation. To give some intuition, we provide concrete examples of $d'$ on permutations.
If $\sigma$ is the identity, then $d'(\sigma) = 0$.
If $\sigma = (1, 2, 3, 4, 6, 5, 7, 8, \ldots)$, $d'(\sigma) = 1$.

If $\sigma = (1, 2, 3, 4, 6, 7, 5, 8, \ldots)$, $d'(\sigma) = 1$. Once $5$ fell out of the top-5, its displacement did not accumulate any further; this may happen when only the top-5 predictions are shown to the user.
If $\sigma = (2, 1, 3, 4, 5, 6, \ldots)$, $d'(\sigma) = 2$.
If $\sigma = (3, 1, 2, 4, 5, 6, \ldots)$, $d'(\sigma) = 4$.
Also, $d'((2, 3, 4, 5, 6, \ldots, 1)) = 5$.
Distinctly, $d'((1, 2, 3, 5, 6, 4, 7, 8, \ldots)) = 2$.
As a final example, $d'((5, 4, 3, 2, 1, 6, 7, 8, 9, \ldots)) = 12$.

It may be that we want perturbation robustness for all predictions, including classes with lesser relevance. In such cases, it is still common that the displacement of the top prediction matters more than the displacement of, say, the 500th ranked class. For this there are many possibilities, such as the measure $d'(\sigma) = \sum_{i=1}^{1000} w_i |w_i - w_{\sigma(i)}|$ such that $w_i = 1/i$. This uses a Zipfian assumption about the rankings of the classes: the first class is $n$ times as relevant as the $n$th class. Other possibilities involve using logarithms rather than hyperbolic functions as in the discounted cumulative gain (Kumar & Vassilvitskii, 2010). One could also use the class probabilities provided by the model (should they exist). However such a measure could make it difficult to compare models since some models tend to be more uncalibrated than others (Guo et al., 2017).

As progress is made on this task, researchers may be interested in perturbations which are more likely to cause unstable predictions. To accomplish that, researchers can simply compare a frame with the frame two frames ahead rather than just one frame ahead. We provide concrete code of this slight change in the metric at https://github.com/hendrycks/robustness. For nontemporal perturbation sequences, i.e., noise sequences, we provide sequences where the noise perturbation is larger.

## D    FULL CORRUPTION ROBUSTNESS RESULTS

IMAGENET-C corruption relative robustness results are in Table 2. Since we use AlexNet errors to normalize Corruption Error values, we now specify the value $\frac{1}{5} \sum_{s=1}^{5} E_{s,\text{Corruption}}^{\text{AlexNet}}$ for each corruption type. Gaussian Noise: 88.6%, Shot Noise: 89.4%, Impulse Noise: 92.3%, Defocus Blur: 82.0%, Glass Blur: 82.6%, Motion Blur: 78.6%, Zoom Blur: 79.8%, Snow: 86.7%, Frost: 82.7%, Fog: 81.9%, Brightness: 56.5%, Contrast: 85.3%, Elastic Transformation: 64.6%, Pixelate: 71.8%, JPEG: 60.7%, Speckle Noise: 84.5%, Gaussian Blur: 78.7%, Spatter: 71.8%, Saturate: 65.8%.

| | | | Noise | | | Blur | | | | Weather | | | | Digital | | | |
|---|---|---|---|---|---|---|---|---|---|---|---|---|---|---|---|---|---|
| Network | Error | **Rel. mCE** | Gauss. | Shot | Impulse | Defocus | Glass | Motion | Zoom | Snow | Frost | Fog | Bright | Contrast | Elastic | Pixel | JPEG |
| AlexNet | 43.5 | 100.0 | 100 | 100 | 100 | 100 | 100 | 100 | 100 | 100 | 100 | 100 | 100 | 100 | 100 | 100 | 100 |
| SqueezeNet | 41.8 | 117.9 | 118 | 116 | 114 | 104 | 110 | 106 | 105 | 106 | 110 | 98 | 101 | 100 | 126 | 129 | 229 |
| VGG-11 | 31.0 | 123.3 | 122 | 121 | 125 | 116 | 129 | 121 | 115 | 114 | 113 | 99 | 86 | 102 | 151 | 161 | 174 |
| VGG-19 | 27.6 | 122.9 | 114 | 117 | 122 | 118 | 136 | 123 | 122 | 114 | 111 | 88 | 82 | 98 | 165 | 161 | 172 |
| VGG-19+BN | 25.8 | 111.1 | 104 | 105 | 114 | 108 | 132 | 114 | 119 | 102 | 100 | 79 | 68 | 89 | 165 | 125 | 144 |
| ResNet-18 | 30.2 | 103.9 | 104 | 106 | 111 | 100 | 116 | 108 | 112 | 103 | 101 | 89 | 67 | 87 | 133 | 97 | 126 |
| ResNet-50 | 23.9 | 105.0 | 104 | 107 | 107 | 97 | 126 | 107 | 110 | 101 | 97 | 79 | 62 | 89 | 146 | 111 | 132 |

Table 2: Clean Error, Relative mCE, and Relative Corruption Errors values of different corruptions and architectures on IMAGENET-C. All models are trained on clean ImageNet images, not IMAGENET-C images. Here "BN" abbreviates Batch Normalization (Ioffe & Szegedy, 2015).

## E    FULL PERTURBATION ROBUSTNESS RESULTS

IMAGENET-P mFR values are in Table 3, and mT5D values are in Table 4. Since we use AlexNet errors to normalize our measures, we now specify the value $\text{FP}_{\text{Perturbation}}^{\text{AlexNet}}$ for each corruption type. Gaussian Noise: 23.65%, Shot Noise: 30.06%, Motion Blur: 9.30%, Zoom Blur: 5.94%, Snow: 11.93%, Brightness: 4.89%, Translate: 11.01%, Rotate: 13.10%, Tilt: 7.05%, Scale: 23.53%, Speckle Noise: 18.65%, Gaussian Blur: 2.78%, Spatter: 5.05%, Shear: 10.66%.

Also, the $\text{uT5D}_{\text{Perturbation}}^{\text{AlexNet}}$ values are as follows. Gaussian Noise: 4.77, Shot Noise: 5.76, Motion Blur: 1.93, Zoom Blur: 1.34, Snow: 2.42, Brightness: 1.19, Translate: 2.63, Rotate: 2.95,

Tilt: 1.75, Scale: 4.48, Speckle Noise: 3.89, Gaussian Blur: 0.70, Spatter: 1.26, Shear: 2.48.

| Network | Error | **mFR** | Noise | | Blur | | Weather | | Digital | | | |
|---|---|---|---|---|---|---|---|---|---|---|---|---|
| | | | Gaussian | Shot | Motion | Zoom | Snow | Bright | Translate | Rotate | Tilt | Scale |
| AlexNet | 43.5 | 100.0 | 100 | 100 | 100 | 100 | 100 | 100 | 100 | 100 | 100 | 100 |
| SqueezeNet | 41.8 | 112.6 | 147 | 140 | 109 | 109 | 105 | 110 | 101 | 103 | 109 | 93 |
| VGG-11 | 31.0 | 74.9 | 89 | 90 | 85 | 84 | 80 | 76 | 52 | 64 | 69 | 59 |
| VGG-19 | 27.6 | 66.9 | 75 | 76 | 82 | 84 | 73 | 74 | 41 | 54 | 60 | 49 |
| VGG-19+BN | 25.8 | 65.1 | 67 | 66 | 75 | 86 | 70 | 72 | 45 | 56 | 56 | 51 |
| ResNet-18 | 30.2 | 72.8 | 74 | 73 | 75 | 85 | 75 | 78 | 63 | 66 | 73 | 66 |
| ResNet-50 | 23.9 | 58.0 | 59 | 58 | 64 | 72 | 63 | 62 | 44 | 52 | 57 | 48 |

Table 3: Flip Rates and the mFR values of different perturbations and architectures on IMAGENET-P. All models are trained on clean ImageNet images, not IMAGENET-P images. Here "BN" abbreviates Batch Normalization.

| Network | Error | **mT5D** | Noise | | Blur | | Weather | | Digital | | | |
|---|---|---|---|---|---|---|---|---|---|---|---|---|
| | | | Gaussian | Shot | Motion | Zoom | Snow | Bright | Translate | Rotate | Tilt | Scale |
| AlexNet | 43.5 | 100.0 | 100 | 100 | 100 | 100 | 100 | 100 | 100 | 100 | 100 | 100 |
| SqueezeNet | 41.8 | 112.9 | 139 | 133 | 109 | 111 | 107 | 112 | 104 | 106 | 111 | 98 |
| VGG-11 | 31.0 | 83.9 | 98 | 97 | 93 | 90 | 87 | 85 | 63 | 75 | 79 | 71 |
| VGG-19 | 27.6 | 78.6 | 89 | 88 | 92 | 93 | 82 | 86 | 53 | 67 | 74 | 62 |
| VGG-19+BN | 25.8 | 80.5 | 85 | 82 | 90 | 97 | 84 | 88 | 61 | 72 | 80 | 66 |
| ResNet-18 | 30.2 | 87.0 | 89 | 87 | 89 | 95 | 88 | 92 | 78 | 82 | 89 | 80 |
| ResNet-50 | 23.9 | 78.3 | 82 | 79 | 84 | 89 | 80 | 84 | 64 | 73 | 80 | 67 |

Table 4: Top-5 Distances and the mT5D values of different perturbations and architectures on IMAGENET-P.

## F  INFORMATIVE ROBUSTNESS ENHANCEMENT ATTEMPTS

**Stability Training.**  Stability training is a technique to improve the robustness of deep networks (Zheng et al., 2016). The method's creators found that training on images corrupted with noise can lead to underfitting, so they instead propose minimizing the cross-entropy from the noisy image's softmax distribution to the softmax of the clean image. The authors evaluated performance on images with subtle differences and suggested that the method provides additional robustness to JPEG corruptions. We fine-tune a ResNet-50 with stability training for five epochs. For training with noisy images, we corrupt images with uniform noise, where the maximum and minimum of the uniform noise is tuned over $\{0.01, 0.05, 0.1\}$, and the stability weight is tuned over $\{0.01, 0.05, 0.1\}$. Across all noise strengths and stability weight combinations, the models with stability training tested on IMAGENET-C have a larger mCEs than the baseline ResNet-50's mCE. Even on unseen noise corruptions, stability training does not increase robustness. However, the perturbation robustness slightly improves. The best model according to the IMAGENET-P validation set has an mFR of 57%, while the original ResNet's mFR is 58%. An upshot of this failure is that benchmarking robustness-enhancing techniques requires a diverse test set.

**Image Denoising.**  An approach orthogonal to modifying model representations is to improve the inputs using image restoration techniques. Although *general* image restoration techniques are not yet mature, denoising restoration techniques are not. We thus attempt restore an image with the denoising technique called non-local means (Buades & Coll, 2005). The amount of denoising applied is determined by the noise estimation technique of Donoho & Johnstone (1993). Therefore clean images receive should nearly no modifications from the restoration method, while noisy images should undergo considerable restoration. We found that denoising increased the mCE from 76.7% to 82.1%. A plausible account is that the non-local means algorithm striped the images of their subtle details even when images lacked noise, despite having the non-local means algorithm governed by the noise estimate. Therefore, the gains in noise robustness were wiped away by subtle blurs to

images with other types of corruptions, showing that targeted image restoration can prove harmful for robustness.

**10-Crop Classification.**   Viewing an object at several different locations may give way to a more stable prediction. Having this intuition in mind, we perform 10-crop classification. 10-crop classification is executed by cropping all four corners and cropping the center of an image. These crops and their horizontal mirrors are processed through a network to produce 10 predicted class probability distributions. We average these distributions to compute the final prediction. Of course, a prediction informed by 10-crops rather than a single central crop is more accurate. Ideally, this revised prediction should be more robust too. However, the gains in mCE do not outpace the gains in accuracy on a ResNet-50. In all, 10-crop classification is a computationally expensive option which contributes to classification accuracy but not noticeably to robustness.

**Smaller Models.**   All else equal, "simpler" models often generalize better, and "simplicity" frequently translates to model size. Accordingly, smaller models may be more robust. We test this hypothesis with CondenseNets (Huang et al., 2017a). A CondenseNet attains its small size via sparse convolutions and pruned filter weights. An off-the-shelf CondenseNet ($C = G = 4$) obtains a 26.3% error rate and a 80.8% mCE. On the whole, this CondenseNet is slightly less robust than larger models of similar accuracy. Even more pruning and sparsification yields a CondenseNet ($C = G = 8$) with both deteriorated performance (28.9% error rate) and robustness (84.6% mCE). Here again robustness is worse than larger model robustness. Though models fashioned for mobile devices are smaller and in some sense simpler, this does not improve robustness.

## G   A SEPARATE TYPE OF ROBUSTNESS

Another goal for machine learning is to learn the fundamental structure of categories. Broad categories, such as "bird," have many subtypes, such as "cardinal" or "bluejay." Humans can observe previously unseen bird species yet still know that they are birds. A test of learned fundamental structure beyond superficial features is *subtype robustness*. In subtype robustness we test generalization to unseen subtypes which share share essential characteristics of a broader type. We repurpose the ImageNet-22K dataset for a closer investigation into subtype robustness.

**Subtype Robustness.**   A natural image dataset with a hierarchical taxonomy and numerous types and subtypes is ImageNet-22K, an ImageNet-1K superset. In this subtype robustness experiment, we manually select 25 broad types from ImageNet-22K, listed in the next paragraph. Each broad type has many subtypes. We call a subtype "seen" if and only if it is in ImageNet-1K and a subtype of one of the 25 broad types. The subtype is "unseen" if and only if it is a subtype of the 25 broad types and is from ImageNet-22K but not ImageNet-1K. In this experiment, the correct classification decision for an image of a subtype is the broad type label. We take pre-trained ImageNet-1K classifiers which have not trained on unseen subtypes. Next we fine-tune the last layer of these pre-trained ImageNet-1K classifiers on *seen* subtypes so that they predict one of 25 broad types. Then, we test the accuracy on images of seen

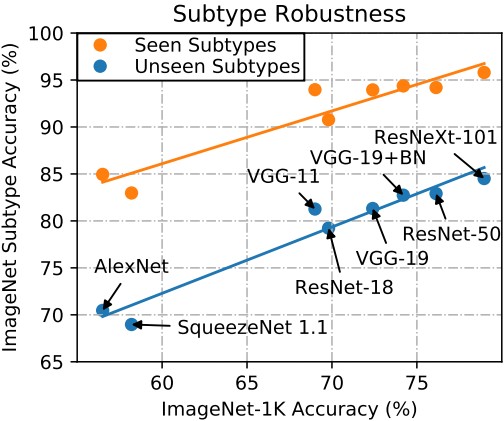

Figure 9: ImageNet classifiers and their robustness to unseen subtypes. Unseen subtypes of known broad types are noticeably harder for classifiers.

subtypes and on images of unseen subtypes. Accuracy on unseen subtypes is our measure of subtype robustness. Seen and unseen accuracies are shown in Figure 9, while the ImageNet-1K classification accuracy before fine-tuning is on the horizontal axis. Despite only having 25 classes and having trained on millions of images, these classifiers demonstrate a subtype robustness performance gap that should be far less pronounced. We also observe that the architectures proposed so far hardly deviate from the trendline.

The 25 broad types which we selected from ImageNet are as follows. Amphibian (n01627424), Appliance (n02729837), Aquatic Mammal (n02062017), Bird (n01503061), Bear (n02131653), Beverage (n07881800), Big cat (n02127808), Building (n02913152), Cat (n02121620), Clothing (n03051540), Dog (n02084071), Electronic Equipment (n03278248), Fish (n02512053), Footwear (n03380867), Fruit (n13134947), Fungus (n12992868), Geological Formation (n09287968), Hoofed Animal (n02370806), Insect (n02159955), Musical Instrument (n03800933), Primate (n02469914), Reptile (n01661091), Utensil (n04516672), Vegetable (n07707451), Vehicle (n04576211).

