# OpenReview forum: "Benchmarking Neural Network Robustness to Common Corruptions and Perturbations"
_ICLR.cc/2019/Conference_

### Official Review · AnonReviewer1 · 2018-11-03
**Exciting paper!**

**Rating:** 9
**Confidence:** 4

**Review:**

Summary: This paper observes that a major flaw in common image-classification networks is their lack of robustness to common corruptions and perturbations. The authors develop and publish two variants of the ImageNet validation dataset, one for corruptions and one for perturbations. They then propose metrics for evaluating several common networks on their new datasets and find that robustness has not improved much from AlexNet to ResNet. They do, however, find several ways to improve performance including using larger networks, using ResNeXt, and using adversarial logit pairing.

Quality: The datasets and metrics are very thoroughly treated, and are the key contribution of the paper. Some questions: What happens if you combine ResNeXt with ALP or histogram equalization? Or any other combinations? Is ALP equally beneficial across all networks? Are there other useful adversarial defenses?

Clarity: The novel validation sets and reasoning for them are well-explained, as are the evaluation metrics. Some explanation of adversarial logit pairing would be welcome, and some intuition (or speculation) as to why it is so effective at improving robustness.

Originality: Although adversarial robustness is a relatively popular subject, I am not aware of any other work presenting datasets of corrupted/perturbed images.

Significance: The paper highlights a significant weakness in many image-classification networks, provides a benchmark, and identifies ways to improve robustness. It would be improved by more thorough testing, but that is less important than the dataset, metrics and basic benchmarking provided.

Question: Why do authors do not recommend training on the new datasets?

---

> ### Author Response · Authors · 2018-11-27
> **Reviewer 1 Reply**
>
> We thank you for your careful analysis of our paper.
>
> “Question: Why do authors do not recommend training on the new datasets?”
> We do not suggest this as the datasets are corrupted or perturbed forms of clean ImageNet validation images, and that training on these specific corruptions would no longer provide a test of generalization ability to novel forms of corruptions. Researchers could train on various other corruptions, such as film grain, adversarial noise, HSV noise, uniform noise, high-pass filtering, median blur, spherical camera distortions, pincushion distortions, out-of-distribution object occlusions, stylized images ( https://openreview.net/forum?id=Bygh9j09KX ), lens scratches, image quilting, color quantization, etc.
>
> “Are there other useful adversarial defenses?”
> Different adversarial training schemes can degrade accuracy so much that they performed worse on these benchmarks. Many other adversarial defenses which do not use train on adversarial or benign noise have been shown not to provide robustness on noise corruptions (see the thorough work of https://openreview.net/pdf?id=S1xoy3CcYX Figure 3). In the coming month, we intend to explore more combinations of techniques to increase robustness, such as the combinations you suggest. In the appendix we explicate four attempts which did not lead to added robustness.

---

### Official Review · AnonReviewer3 · 2018-11-12
**An important benchmark for measuring the robustness of computer vision models**

**Rating:** 9
**Confidence:** 5

**Review:**

This paper introduces new benchmarks for measuring the robustness of computer vision models to various image corruptions. In contrast with the popular notion of “adversarial robustness”, instead of measuring robustness to small, worst-case perturbations this benchmark measures robustness in the average case, where the corruptions are larger and more likely to be encountered at deployment time. The first benchmark “Imagenet-C” consists of 15 commonly occurring image corruptions, ranging from additive noise, simulated weather corruptions, to digital corruptions arising from compression artifacts. Each corruption type has several levels of severity and overall corruption score is measured by improved robustness over a baseline model (in this case AlexNet). The second benchmark “Imagenet-P” measures the consistency of model predictions in a sequence of slightly perturbed image frames. These image sequences are produced by gradually varying an image corruption (e.g. gradually blurring an image). The stability of model predictions is measured by changes in the order of the top-5 predictions of the model. More stable models should not change their prediction to minute distortions in the image. Extensive experiments are run to benchmark recent architecture developments on this new benchmark. It’s found that more recent architectures are more robust on this benchmark, although this gained robustness is largely due to the architectures being more accurate overall. Some techniques for increasing model robustness are explored, including a recent adversarial defense “Adversarial Logit Pairing”, this method was shown to greatly increase robustness on the proposed benchmark. The authors recommend future work benchmark performance on this suite of common corruptions without training on this corruptions directly, and cite prior work which has found that training on one corruption type typically does not generalize to other corruption types. Thus the benchmark is a method for measuring model performance to “unknown” corruptions which should be expected during test time.

In my opinion this is an important contribution which could change how we measure the robustness of our models. Adversarial robustness is a closely related and popular metric but it is extremely difficult to measure and reported values of adversarial robustness are continuously being falsified [1,2,3]. In contrast, this benchmark provides a standardized and computationally tractable benchmark for measuring the robustness of neural networks to image corruptions. The proposed image corruptions are also more realistic, and better model the types of corruptions computer vision models are likely to encounter during deployment. I hope that future papers will consider this benchmark when measuring and improving neural network robustness. It remains to be seen how difficult the proposed benchmark will be, but the authors perform experiments on a number of baselines and show that it is non-trivial and interesting. At a minimum, solving this benchmark is a necessary step towards robust vision classifiers.

Although I agree with the author’s recommendation that future works not train on all of the Imagenet-C corruptions, I think it might be more realistic to allow training on a subset of the corruptions. The reason why I mention this is it’s unclear whether or not adversarial training should be considered as performing data augmentation on some of these corruptions, it certainly is doing some form of data augmentation. Concurrent work [4] has run experiments on a resnet-50 for Imagenet and found that Gaussian data augmentation with large enough sigma (e.g. sigma = .4 when image pixels are on a [0,1] scale) does improve robustness to pepper noise and Gaussian blurring, with improvements comparable to that of adversarial training. Have the authors tried Gaussian data augmentation to see if it improves robustness to the other corruptions? I think this is an important baseline to compare with adversarial training or ALP.

Few specific comments/typos:

Page 2 “l infinity perturbations on small images”

The (Stone, 1982) reference is interesting, but it’s not clear to me that their main result has implications for adversarial robustness. Can the authors clarify how to map the L_p norm in function space of ||T_n - T(theta) || to the traditional notion of adversarial robustness?

1. https://arxiv.org/pdf/1705.07263.pdf
2. https://arxiv.org/pdf/1802.00420.pdf
3. https://arxiv.org/pdf/1607.04311.pdf
4. https://openreview.net/forum?id=S1xoy3CcYX&noteId=BklKxJBF57

---

> ### Author Response · Authors · 2018-11-27
> **Reviewer 3 Reply**
>
> Thank you for your interest in this topic and your analysis of our paper.
>
> “I think it might be more realistic to allow training on a subset of the corruptions.”
> Researchers could train on various other corruptions, such as film grain, adversarial noise, HSV noise, uniform noise,
> high-pass filtering, median blur, spherical camera distortions, pincushion distortions, out-of-distribution object occlusions, stylized images ( https://openreview.net/forum?id=Bygh9j09KX ), lens scratches, image quilting, color quantization, etc. We have updated the text to make it clearer that researchers can train on more than just cropped and flipped images, but we still do not want researchers training on the test corruptions. In the paper we experimented with uniform noise data augmentation in the stability training experiment and found minor perturbation robustness gains, but not with Gaussian noise with a large standard deviation.
>
> Thank you for pointing out that the brief Stone comment requires much more context. For that reason we have removed the citation. Essentially, if f is a model and f^\hat is an approximation, and if input x is d-dimensional, then if we want | f(x) - f^\hat (x) | < epsilon, then in some scenarios the number of samples necessary is ~ epsilon^{-d}. Other context is on slide 10 of https://github.com/joanbruna/MathsDL-spring18/blob/master/lectures/lecture1.pdf
>
> “l infinity perturbations on small images”
> Thanks to your suggestion, we have changed this to “perturbations on small images.” We kept the word “small” as the images often have side length 32 pixels. We removed “l_infinity” since that method has had some success for perturbations which are small in an l_2 sense.

---

### Official Review · AnonReviewer2 · 2018-11-14
**It is an importance work for deep learning research.**

**Rating:** 7
**Confidence:** 3

**Review:**

This paper introduces two benchmarks for image classifier robustness, ImageNet-C and Image-P. The benchmarks cover two important cases in classifier robustness  which are ignored by most current researchers. The authors' evaluations also show that current deep learning methods have wide room for improvement. To our best knowledge, this is the first work that provides systematically a common benchmarks for the deep learning community.  The reviewer believes that these two benchmarks can play an important role in the research of image classifier robustness.

---

> ### Author Response · Authors · 2018-11-27
> **Reviewer 2 Reply**
>
> We thank you for taking time to review our work.

---

### Public Comment · (anonymous) · 2018-11-09
**Interesting work!**

You've shown that ALP performs so well on this benchmark, but ALP performs some form of data augmentation by training on worst-case perturbations. Therefore, its unclear whether or not this satisfies the recommendation that future work not train on the Imagenet-C corruptions. Have you compared the ALP model with simply performing Gaussian data augmentation? Some recent adversarial defense works have reported that Gaussian data augmentation improves small perturbation robustness.

---

> ### Author Response · Authors · 2018-11-27
> **Augmentation Clarification**
>
> Noises such as those from gradients or uniform noise are perfectly acceptable forms of augmentation for this task. In the stability training experiment, we observed only minor gains in perturbation robustness when training with uniform noise, but perhaps training with more severe uniform noise could improve corruption robustness. In the revised paper, we make it clearer that training with other forms of data augmentation is acceptable. Please forgive this confusion.

---

### Public Comment · ~Dogancan_Temel1 · 2018-11-23
**Related Work**

I would like to thank the authors for focusing on such a critical issue in a comprehensive manner. Algorithmic solutions behind the core technologies have to be robust even under challenging conditions in order for such technologies to be effective and useful in our daily lives. With more and more studies similar to the submitted ICLR work, we can identify the weaknesses and strengths of existing algorithms to develop more reliable perception systems.  One of the main contributions of the submitted work is based on the common corruptions and perturbations not worst-case adversarial perturbations. With a similar mindset, we have introduced three datasets, two for traffic signs (CURE-TSR [2], CURE-TSD [3]) and one for generic objects (CURE-OR [1]) to investigate the robustness of recognition/detection systems under challenging conditions corresponding to adversaries that can naturally occur in real-world environments and systems. The controlled challenging conditions in the CURE-OR [1] dataset include underexposure, overexposure, blur, contrast, dirty lens, image noise, resizing, and loss of color information. And the controlled conditions in the CURE-TSR [2] and CURE-TSD [3] datasets include rain, snow, haze, shadow, underexposure, overexposure, blur, dirtiness, loss of color information, sensor and codec errors.  Based on the similarities between introduced datasets and conducted studies, including aforementioned studies in the literature analysis of the submitted paper can be helpful to reflect recent related work. Looking forward to authors’ upcoming studies, thanks.

[1] D. Temel*, J. Lee*, and G. AlRegib, “CURE-OR: Challenging unreal and real environments for object recognition,” IEEE International Conference on Machine Learning and Applications, Orlando, Florida, USA, December 2018, (*: equal contribution). https://arxiv.org/abs/1810.08293
[2] D. Temel, G. Kwon*, M. Prabhushankar*, and G. AlRegib, “CURE-TSR: Challenging unreal and real environments for traffic sign recognition,” Advances in Neural Information Processing Systems (NIPS) Workshop on Machine Learning for Intelligent Transportation Systems, Long Beach, U.S., December 2017, (*: equal contribution).https://arxiv.org/abs/1712.02463
[3] D. Temel and G. AlRegib, “Traffic Signs in the Wild: Highlights from the IEEE Video and Image Processing Cup 2017 Student Competition [SP Competitions],” in IEEE Signal Processing Magazine, vol. 35, no. 2, pp. 154-161, March 2018.https://arxiv.org/abs/1810.06169

---

> ### Author Response · Authors · 2018-11-27
> **Cited Work**
>
> Thank you for your interest in this topic and making us aware of your work. An earlier draft of our work appeared months before the time of the ICLR submission deadline, and we have added all citations to your traffic sign recognition work and your parallel works.

---

> > ### Public Comment · ~Dogancan_Temel1 · 2018-11-29
> > **Thanks**
> >
> > Thanks for the quick response. Also, I really appreciated the additional section at the end of the paper where you talk about the robustness enhancement attempts, it is good to know not just what worked but also what did not work and why.

---

### Author Response · Authors · 2018-11-27
**Minor Revision Posted**

We should like to thank all of the reviewers and commenters for their constructive comments and kind reception. Independent from their comments, we have created CIFAR-10-C and CIFAR-10-P which could be adequate for rapid experimentation. Also in the revised version is a new appendix where we briefly analyze a different notion of robustness separate from our main contributions. We will respond to each reviewer’s comments individually.

---

### Public Comment · (anonymous) · 2018-11-29
**Discussion of prior work missing**

I would like to point out that this submission is missing a discussion of some very relevant prior work. That work already evaluates the robustness of ML classifiers to naturally occurring transformations such as rotations and translations. Specifically:

• Fawzi et al. (2015) [https://arxiv.org/abs/1507.06535] compute the minimum transformation (composed of rotations, translations, scaling, etc.) needed to cause a misclassification for a wide variety of models. They find that it is relatively small, and make several observations about the relative robustness of different classifiers.

• Engstrom et al. (2017) [https://arxiv.org/abs/1712.02779] fix a range of rotations and translations and compute that worst-case accuracy of models over this space. They also find models to be relatively non-robust and propose methods for improving it.

• Kanbak et al. (2018) [https://arxiv.org/abs/1711.09115] develop a first-order method to find such worst-case transformations fast. They show that this method can then be used to perform adversarial training and improve the model's robustness.

```The authors were already notified about existence of some of this prior work a long time ago, but still seem to dismiss it.

---

> ### Author Response · Authors · 2018-11-30
> **Revised Discussion**
>
> We would be happy to expand the related works further in future revisions of this draft. We have cited the sender of the e-mail from "a long time ago" twice in the current draft, but we can add more in a future revision.

---

> > ### Comment · Area_Chair1 · 2018-12-04
> > **general approach to anonymous remarks about missing citations**
> >
> > It sounds as if the authors agree with the suggestion, although I am not completely sure. However, I would like to emphasize that if they did not agree, then it would be up to the reviewers to determine whether adding these citations was important. Without investigating further, I have no position either way.
> >
> > But, in general, our obligation as scientists is to cite other work when doing so benefits the reader. We should exercise our own taste in what we cite and avoid citing things that we do not think enhance the experience of the reader.
> >
> > Authors: please don't hesitate to ask reviewers+AC to weigh-in if you are even in doubt about the importance of adding a particular citation.

---

> > > ### Public Comment · (anonymous) · 2018-12-04
> > > **response**
> > >
> > > If I am interpreting the comment correctly, the author seems to be saying that they have cited the sender of the email's other papers, but does not see the need to cite any of the papers listed above.
> > >
> > > This is a bit confusing as our comments are not an attempt to "extort" citations, but rather an effort to put this work in the right context. The fact that the authors cite other (admittedly less relevant) papers of the email sender does not render the suggested work less relevant.
> > >
> > > To reiterate, I believe that all these works are very relevant to the subject of the above paper. If the authors do not want to cite these papers, that is okay - however one would expect them to at least explain in OpenReview why or give a brief comparison.

---

> > > > ### Comment · AnonReviewer3 · 2018-12-13
> > > > **Unclear to me what these papers add over currently cited papers**
> > > >
> > > > This work already cites many previous fragility studies, both from robustness to random corruptions/perturbations and with respect to worst-case corruptions, some works which include robustness to translations. Based on a quick reading of the proposed additional citations it is unclear to me what these works add on top of what is already cited. I have no strong opinion either way whether additional citations are added, I leave it up to the authors or other reviewers to decide what is best for the proper context of this work.

---

> > > ### Author Response · Authors · 2018-12-11
> > > **Citations**
> > >
> > > Please excuse the delayed response, as we were at NeurIPS.
> > >
> > > The original poster sent an e-mail many months ago including numerous links to many papers, including several of their own. We conclude this because we received only one e-mail with citation suggestions. In consequence, we cited two of the papers authored by the person sending the e-mail, giving a sentence description for each citation. Several months later, the email sender posted the comment above. The only link which appeared in both the e-mail and in the comment above is Engstrom et al. (which is under review). The Fawzi et al. and Kanbak et al. papers are new to us. These may be added to the "ConvNet Fragility Studies" section. We think it is a reasonable suggestion to spend more time discussing other ConvNet perturbation fragility findings, although we do already cite works which mention translation instability (such as the parallel work of Azulay & Weiss, 2018).

---

### Author Response · Authors · 2018-11-30
**Data Augmentation with Stylized ImageNet Improves Corruption Robustness**

A parallel submission proposes to train classifiers on stylized ImageNet images. The aim is to make classifiers rely less on texture and more on shape. https://openreview.net/pdf?id=Bygh9j09KX

We have found that this method indeed improves corruption robustness. A ResNet-50 obtains an mCE of 76.70%, while a ResNet-50 trained on both ImageNet images and stylized ImageNet images has an mCE of 69.32% (with general improvements noise, blur, weather, and digital categories).

---

### Public Comment · (anonymous) · 2018-12-16
**Question about the Representativity and the Time-Efficiency of the benchmarks**

Hi, it’s an interesting work!

I would like to ask the authors how to ensure that the benchmarks are sufficiently representative to evaluate the robustness of models.

Will the benchmarks be updated in the future as new adversarial attacks (Corruptions or Perturbations) emerge?

---

> ### Author Response · Authors · 2018-12-19
> **Reply**
>
> Thank you for your interest! Since this task is not adversarial in nature, we do not intend to continually modify the corruptions to subvert new approaches, much like how CIFAR-10 did not continually change to make classification harder for every new architecture and method. Improved generalization to unseen corruptions suggests improved corruption robustness. However if necessary we are open to updating the benchmark, but we will first see whether the research community experiments in this setting.

---

### Meta-Review · Area_Chair1 · 2018-12-13
**clear consensus to accept this paper**

**Confidence:** 5
**Recommendation:** Accept (Poster)

**Metareview:**

The reviewers have all recommended accepting this paper thus I am as well. Based on the reviews and the selectivity of the single track for oral presentations, I am only recommending acceptance as a poster.